# Comparative Proteomics of Peritrophic Matrix Provides an Insight into its Role in Cry1Ac Resistance of Cotton Bollworm *Helicoverpa armigera*

**DOI:** 10.3390/toxins11020092

**Published:** 2019-02-02

**Authors:** Minghui Jin, Chongyu Liao, Swapan Chakrabarty, Kongming Wu, Yutao Xiao

**Affiliations:** 1Agricultural Genomics Institute at Shenzhen, Chinese Academy of Agricultural Sciences, Shenzhen 518120, China; jinminghui@163.com (M.J.); leochongyu@163.com (C.L.); swapan.ag.sau@gmail.com (S.C.); 2The State Key Laboratory for Biology of Plant Disease and Insect Pests, Institute of Plant Protection, Chinese Academy of Agricultural Sciences, West Yuanmingyuan Road, Beijing, 100193, China; wukongming@caas.cn

**Keywords:** *Helicoverpa armigera*, Bt, midgut, transcriptome, peritrophic matrix, proteome

## Abstract

Crystalline (Cry) proteins from *Bacillus thuringiensis* (Bt) are widely used in sprays and transgenic crops to control insect pests, but the evolution of insect resistance threatens their long-term use. Different resistance mechanisms have been identified, but some have not been completely elucidated. Here, the transcriptome of the midgut and proteome of the peritrophic matrix (PM) were comparatively analyzed to identify potential mechanism of resistance to Cry1Ac in laboratory-selected strain XJ10 of *Helicoverpa armigera*. This strain had a 146-fold resistance to Cry1Ac protoxin and 45-fold resistance to Cry1Ac activated toxin compared with XJ strain. The mRNA and protein levels for several trypsin genes were downregulated in XJ10 compared to the susceptible strain XJ. Furthermore, 215 proteins of the PM were identified, and nearly all had corresponding mRNAs in the midgut. These results provide new insights that the PM may participate in Bt resistance.

## 1. Introduction

The toxins produced by *Bacillus thuringiensis* (Bt) have been used widely in transgenic plants for pest control with little or no harm to people and most non-target organisms [1,2,3,4]. The cotton bollworm, *Helicoverpa armigera*, is a principal cotton pest and inflicts major losses worldwide [5,6,7]. Transgenic cotton lines that produce Bt toxin have been useful in suppressing this polyphagous pest [6], but they often develop resistance to the toxins after long-term use [8,9,10].

Understanding the mode of action and the mechanisms of resistance to Bt proteins can help to enhance and sustain their efficacy against pests. The mode of action of Bt toxicity is complex. Models of the Bt mode of action agree that protoxins (the full-length forms of Cry1Ac proteins) are digested by midgut proteases into activated toxins, which then bind to insect midgut receptors, forming lytic pores in the membrane and leading to cell breakdown [11,12]. During this activation, five hundred amino acids from the carboxyl terminus and 40 amino acids from the amino terminus were removed. The relative molecular mass of protoxins converted from 130 kDa to 55 ~ 65 kDa of activated toxins [12,13]. It has been reported that reduced conversion of protoxin can cause greater resistance to protoxins than activated toxins [14,15,16,17,18].

Blocking any of these steps will lead to resistance. To date, numerous Bt-resistant insect populations have been selected in the laboratory [14,19,20,21,22,23], and various resistance mechanisms have been identified, such as altered activation of midgut digestive proteases, toxin sequestration by glycolipid moieties or esterase, elevated immune response, and reduced binding of Cry toxins [2,11,24,25].

The peritrophic matrix (PM), the acellular, porous tubular lining of the arthropod gut, is secreted by most insects and important for digestion, serving as a semi-permeable barrier between the epithelial cells of gut and the food bolus and protecting the midgut epithelium from infection by pathogens, damage by toxins, and mechanical damage by rough food particles [26,27,28,29]. The PM is composed of chitin fibrils with associated proteoglycans and glycoproteins and is proposed to assist the digestion process and immobilization of digestive enzymes, allowing reuse of hydrolytic enzymes and efficient acquisition of nutrients. Furthermore, the PM may also be a valid target for insect control [30,31,32].

Our previous work with 10 laboratory-selected strains of *H. armigera* suggested that reduced protease activity is associated with resistance to Cry1Ac [33]. Here we evaluated a resistant strain XJ10 (derived from XJ5) and found that the resistance ratio of Cry1Ac protoxin is much higher than that of the activated Cry1Ac toxin. We hypothesized that this ratio may be associated with the gut digestive proteases that take part in the conversion of the protoxin. In a comparative analysis of transcriptomes (RNA-seq) of midguts and proteomes (iTRAQ) of peritrophic matrix between XJ strain and XJ10 strain, we aimed to identify proteins that may be associated with Bt resistance and in the PM of *H. armigera*.

## 2. Results

### 2.1. Resistance to Cry1Ac Protoxin and Activated Toxin

Bioassay results indicated that the laboratory-selected XJ10 strain of *H. armigera* was resistant to Cry1Ac protoxin and active toxin compared to its parental strain (XJ, Table 1). The resistant ratios were calculated as the concentration (ng Cry1Ac per g diet) of Cry1Ac killing 50% (LC_50_) larvae for XJ10 divided by the LC_50_ for XJ larvae which were 146.8 and 45.0 for protoxin and activated toxin, respectively (Table 1).

### 2.2. Transcriptomic Analysis of Midguts from XJ and XJ10 Larvae

The mRNA transcript levels for genes from the midguts of strain XJ were compared to those from XJ10 strain for differential transcription. The mapping data analysis revealed 10,495 (75.85%) and 10,617 (76.73%) coding genes in the reference genome, respectively. We used a fold-change ≥ 1.5 and *P*-value < 0.05 as the threshold to judge that differences in expression were significant. We found that nearly 7.9% of all detected genes (457 up- and 378 down-regulated) were differentially expressed between XJ10 and XJ strain (Fig. 1A). To better understand the functional categories that differed between XJ10 and XJ strain, we used Blast2GO to assign GO categories to the 835 DEGs. The distribution of the GO terms are shown in Figure 1B. Cellular process, metabolic process and single-organism process were the major categories annotated under biological process. Cell, cell part and organelle were the major categories annotated under cellular component. As for molecular function, the major categories were binding and catalytic activity. Furthermore, KEGG analysis showed that 15 pathways were substantially enriched (*P* < 0.05), including ribosome, metabolic pathways (Appendix A).

Trypsin and chymotrypsin are very important digestive enzymes and play important roles in Bt protoxin activation. In this study, as shown in Table 2, 11 trypsin and chymotrypsin were found differentially expressed between XJ10 and XJ. Mechanism of action of Bt toxins is complex, blocking any step may lead to resistance. Besides trypsin, other differentially regulated genes, including ABCs, polycalin, APN and ALP, possibly linked to Bt resistance are also listed in Table 2. We chose Bt resistance-related genes for proteins such as trypsin, esterase, ABCs and Bt receptors for qRT-PCR analysis, which indicated that most of the Bt resistance-related genes had expression patterns similar to those shown by the RNA-seq data (Figure 2).

### 2.3. Proteomic Analysis of PM from XJ and XJ10 Larvae

In the parallel iTRAQ analysis to compare the proteome of PM from XJ and XJ10 (three biological replicates in each group), 215 proteins were identified and quantified, (see details on the proteins in Appendix A). Based on the specific functions of the proteins, we divided them into categories (Table 3) that included chitin-related, digestion-related, lipocalins-related, immune-related and so on. Polycalin and APNs, which have been reported as Cry1Ac receptors, were also found in this study. Similar to other findings, most of the identified proteins could not be characterized or had unknown functions [34,35].

In the GO analysis, the 215 identified proteins were enriched in 16 biological process terms. The top 10 processes (Figure 3A) indicate that most proteins are involved in biological processes related to metabolic processes. Catalytic activity and binding were the most abundant molecular function categories (Figure 3B). Additionally, among 12 other cellular component terms, cell part component had the largest group of proteins (Figure 3C). The results of the GO enrichment revealed that the proteins of PM were predominantly binding proteins and have catalytic activity, located in membrane and involved in metabolic processes.

In the analysis of global changes of PM proteins between strain XJ10 and XJ, 12 proteins were classified as differentially expressed proteins (DEPs) (Table 4). Among these DEPs, most downregulated proteins were active digestive hydrolases including five trypsins, two chymotrypsins, carboxypeptidase A and one uncharacterized protein. In XJ10 strain, α-amylase, which is involved in food digestion, was upregulated. Chitin deacetylase and unconventional myosin were also upregulated in XJ10 strain.

### 2.4. Correlation between Transcriptome and Proteome

Because PM proteins are secreted from the midgut, we analyzed the correlation between the identified proteins of PM and mRNAs of the midgut. The distributions of the ratio of the corresponding protein to the mRNA are shown in Figure 4. Of the 215 identified proteins, 95.8% (205/215) had corresponding mRNAs, indicating that nearly all PM proteins have a corresponding mRNA in the midgut and confirmed that PM proteins are secreted from the midgut. As for the 12 DEPs, their corresponding mRNA expression levels are given in Table 4. For the downregulated DEPs, including four trypsins (XM_021337877.1, XM_021337869.1, XM_021340592.1, and XM_021340602.1) and one uncharacterized protein (XM_021340037.1), the corresponding transcript levels were also down-regulated.

## 3. Discussion

The laboratory-selected Cry1Ac-resistant *H. armigera* strain XJ10 had a resistance ratio of 45 for the active Cry1Ac toxin and 146 for Cry1Ac protoxin. Because the resistance ratio for the Cry1Ac protoxin was much higher than for the activated toxin, these results suggest that reduced activation of Cry1Ac may contribute to resistance in XJ10 strain. However, the resistance ratio for the activated toxin indicated the presence of another resistance mechanism in XJ10. In the present study of the transcriptome of the midgut and proteome of the PM between XJ and XJ10 strains using RNA-seq, hundreds of genes were differently expressed, and 12 of the 215 proteins that were identified by iTRAQ varied in abundance between XJ and XJ10 strain, and nearly all the 215 identified proteins in the PM had corresponding mRNAs in the midgut.

### 3.1. Identification of Potential Resistance Mechanisms in XJ10

The mechanisms of Bt resistance include diverse process, because it could occur by blocking of any of the steps noted above. Among the differentially regulated genes for proteins potentially related to Bt resistance (Table 2 and Table 4), gut proteases play an important role in protoxin activation and in the Bt-resistance mechanism in several lepidopteran species. For example, downregulation of protease gene confers Cry1Ac resistance to *H. armigera* stain Akola-R from India and strain LF5 from China [7,14]. In contrast, increased activity of gut proteases in *Spodoptera littoralis* enhances the resistance to Cry1C, possibly due to overdegradation of the toxin [36]. In the present study, many trypsin genes were differentially expressed in transcriptomic profiles and also in proteome profiles of the PM. The PM is proposed to assist the digestion process by partitioning the gut lumen into ectoperitrophic space (between PM and lumen) and endoperitrophic space (between epithelium and PM) and immobilization of digestive enzymes [36]. These immobilized digestive enzymes may play important roles in the activation of Cry1Ac protoxin. Protease-mediated resistance was first demonstrated in a laboratory-selected strain of *Plodia interpunctella*, for which larval survival was genetically linked with a lack of major trypsin-like gut proteases after Cry1Ac treatment [16]. In other insects, such as *Heliothis virescens* and *Ostrinia nubilalis*, reduced protease activity is associated with resistance to Cry1A toxins [37,38].

Sequestration of the toxin by esterases has also been reported to be associated with resistance. Esterase could be responsible for sequestering large quantities of Cry1Ac in the 275-fold resistant *H. armigera* strain [39]. Here we found that the expression of esterase was upregulated in strain XJ10. Cry toxins are known to have different receptors such as cadherin (Cad), aminopeptidase-N (APN), alkaline phosphatase (ALP) and ABC transporters, which bind to Cry1Ac toxin through a combination of different modes and are important in the receptor-mediated toxicity of Cry toxins. In the case of XJ10, ALP2 and APN1 were upregulated in mRNA level compared with XJ strain. Many studies showed that the mutation of APN1 or downregulation of APN1 or ALP2 is correlated with Bt resistance, but no studies had reported that the upregulated APN1 or ALP2 in mRNA level was related to Bt resistance [40,41,42,43]. APN and ALP are attached to the cell membrane via a glycosyl phosphotidylinositol (GPI) anchor, and the N- and/or O-linked glycans are thought to mediate the binding of Cry toxins. Whether the upregulated expression of ALP2 and APN1 is associated with resistance or a post-translational modification needs further study.

### 3.2. Identification of PM Proteins

In our comprehensive analysis of cotton bollworm larval PM proteins using iTRAQ, we identified 215 proteins, more than previously reported [29], among several major classes of proteins (Table 3), and thus potentially different functions for the PM of the midgut. In this study, the two mucins and the chitin deacetylase (CDA) protein represented two classes of chitin-binding proteins. CDA is a hydrolytic enzyme that degrades the glycosidic bonds of chitin and may control the rigidity and porosity of chitin-containing PM [44,45]. In XJ10, the protein level of CDA was higher than in XJ. Thus, chitin rigidity in XJ10 may be higher and obstruct Cry1Ac toxin entry into the PM.

Forty-six of the identified proteins were digestive hydrolases of various types, including trypsin, carboxypeptidase, α-amylase, serine protease and lipase. Since the midgut is for digestion and absorption, these digestive enzymes might be immobilized on the PM and present in two forms—either bound to the PM or soluble in the gut lumen. The PM may accelerate digestion in cotton bollworm larvae via PM-bound digestive enzymes. In strain XJ10, the level of α-amylase 2-like was higher than in XJ. α-Amylase is widely distributed among animals, plants and microbes and catalyzes the hydrolysis of starch [46], and may have no association with resistance. Of the 12 proteins we found to be differentially expressed, seven were trypsin, which were all downregulated in XJ10. 

Decreased protoxin activation of Cry1Ac or Cry1Ab has been associated with resistance in many insects. Our finding that the resistance ratio of XJ10 for Cry1Ac protoxin was much higher than the resistance ratio of the activated Cry1Ac toxin is similar to previous work in a laboratory-selected strain LF5 of *H. armigera*, for which the resistance ratio was 2.8 times higher for the Cry1Ac protoxin than the activated toxin. The resistance of LF5 is genetically linked with a trypsin gene *HaTryR* and reducing the expression of this gene using RNAi increases the survival of susceptible larvae treated with Cry1Ac protoxin [14], implying that trypsin protease is important for activating the Cry1Ac protoxin. We thus suggest that these downregulated trypsins found in the PM may be responsible for the reduced Cry1Ac activation in XJ10.

Serine proteases are important proteolytic enzymes for digestion and insect innate immune systems [47,48]. Although we identified seven serine proteases and two serine protease inhibitors in this study, their protein expression levels did not change. We also identified some heat shock proteins and numerous proteins in the PM with unknown functions. These proteins should not be neglected, because they may play important roles in the molecular architecture of PM. 

In our exploration of the gene expression changes in the midgut and protein changes in the PM between XJ10 and XJ strain, both the mRNA and protein levels for several trypsins were downregulated, revealing a potential association with Cry1Ac protoxin activation and the high resistance to Cry1Ac protoxin. However, reduced activation cannot explain the 45-fold resistance ratio for the activated Cry1Ac toxin, indicating that another mechanism(s) also contributes to resistance in XJ10. Furthermore, the identification of proteins of PM enabled us to better understand the nature of the PM and its involvement in digestion and deactivation of toxins such as Bt toxins.

## 4. Materials and Methods

### 4.1. Insect Strains

The XJ strain was collected in a cotton field in Xiajin (Shandong Province, China) in 2004 and reared in the laboratory without exposure to Bt toxins or insecticides [33]. The resistant XJ10 strain was derived from strain XJ via selection on Cry1Ac-contaminated artificial diet through a series of progressively more resistant strains, XJ1, XJ5 and XJ10, with the number of each resistant strain corresponding to the concentration of Cry1Ac. Thus, XJ10 was selected on 10 μg Cry1Ac protoxin per ml of diet. During the selection, neonates were reared on the artificial diet with the respective dose of toxin for 7 days, then well-developed larvae were transferred to non-Bt artificial diet until moth emergence. Insects were reared in an insect chamber with a controlled environment (27 ± 2 °C, 75 ± 10% RH, 14L: 10D).

### 4.2. Insect Bioassays

Bioassays on artificial diets with various concentrations of Cry1Ac protoxin or active toxin in 24-well were performed according to the methods of Liu et al. [14]: a 4-day-old larva was placed in each well, with 24 larvae for each treatment. Mortality was recorded after 7 d. The concentration that killed 50% of larvae (LC50) was determined using a probit analysis.

### 4.3. RNA Extraction and cDNA Library Construction

Fifth instar larvae form XJ and XJ10 strains were anesthetized on ice and dissected longitudinally to obtain midguts (n=24, 3 replicates). Total RNA were isolated using Trizol (Invitrogen, USA) according to the manufacturer’s protocol. 1% agarose gels were used to check RNA degradation and contamination and Nanophotometer1 spectrophotometer (IMPLEN, CA, USA) was used to check the purity of RNA. Library construction and sequencing using Illumina HiSeq^TM^ 2500 were done by Gene denovo Biotechnology Co., Ltd., Guangzhou, China. Briefly, mRNA was purified from total RNA using poly T oligo-attached magnetic beads. 

### 4.4. Data Processing and Analysis

Clean reads were obtained by removing reads containing adapters, poly N, and low-quality reads form the raw reads. The clean data were then mapped to the reference genome (Accession NO., PRJNA388211) by TopHat 2 (v 2.0.3.12) [49] using the following modification from default parameters: the distance between mate-pair reads, 50bp; the error of distance between mate-pair reads, ± 80bp; up to two mismatches allowed. The expression levels of genes were estimated using RSEM [50] and normalized using the FPKM (fragments per kb of transcript per million mapped reads). Differential expression analysis was performed using the RStudio with package edgeR. The resulting *P* values were adjusted using the Benjamini and Hochberg’s approach for controlling the false discovery rate. The genes were considered differentially expressed if they had an adjusted *P*-value <0.05 and a change of at least 1.5-fold. Gene Ontology (GO) analysis was conducted for the functional classification of differentially expressed genes (DEGs), and pathway analysis was carried out using Kyoto Encyclopedia of Genes and Genomes (KEGG).

### 4.5. qRT-PCR Analysis of Gene Expression Levels

Total RNA was extracted from the midgut as described above with three replications of samples, then 1 μg of RNA was reverse transcribed for first-strand cDNA synthesis using *TransScript* One-step gDNA Removal and cDNA Synthesis SuperMix (Transgen Biotech, China). Primers (Appendix A) were designed with Beacon Designer 7.9 software (PREMIER Biosoft International, CA, USA). The reaction mixture contained 10 μL TransStart Tip Green qRCR SuperMix (Transgen Biotech, China), 1.0 μL cDNA, 8.2 μL ddH2O, and 0.4 μL each of the forward and reverse gene-specific primers and run in a Stepone Plus Real-time system with thermal cycling set at 95 °C for 3 min; 40 cycles of 95 °C for 10 s and 55 °C for 60 s. The qRT-PCR for each sample was conducted with three technical replicates and three biological replicates. *Actin* was used as a reference gene to normalize the content of cDNA. Relative expression levels for target genes, in relation to the most reliable reference gene, were calculated via the 2^−∆∆CT^ method. Levels were analyzed for significant differences using Student’s *t*-test (SPSS).

### 4.6. iTRAQ-Based Proteome Determination and Data Analysis

The peritrophic matrix (PM) was excised from 5th instars of strains XJ and XJ10 that had been actively feeding, as described previously [29]. Total protein extracted using Tissue Protein Extraction Kit (Cwbiotech Co. Ltd., Beijing, China) following the manufacturer’s method. For digestion, 100 μg of total protein, quantified by the Bradford method, was incubated overnight with Trypsin Gold (Promega; protein: trypsin = 30:1) at 30 °C for 16 h. After trypsin digestion, the resultant peptides were dried by vacuum centrifugation. iTRAQ labeling was performed using an iTRAQ reagent 8PLEX Multiplex kit (Applied Biosystems, US) according to the manufacturer’s protocol. Samples from XJ and XJ10 were labeled with iTRAQ tags 115 and 119, respectively. Then, the pooled mixtures of iTRAQ-labeled peptides were fractionated by strong cationic exchange (SCX) chromatography using a Shimadzu LC-20AB HPLC Pump system. Collected fractions were pooled into 10 final fractions for nano LC-MS/MS analysis after desalting by Strata XC18 column (Phenomenex) and vacuum dried. The MS/MS analysis was performed using a LTQ-Qrbitrap velos mass spectrometer (ThermoFisher Scientific, Rockford, IL, USA). MS data were acquired using a data-dependent top 10 method, dynamically choosing the most abundant precursor ions from the survey scan (300–1800 *m/z*) for HCD fragmentation. Determination of the target value based on predictive automatic gain control. The raw data files were converted into MGF format using Proteome Discover 1.2 (Thermo Fisher, Waltham, MA, USA). Proteins were identified and quantified using the MASCOT engine (Matrix Science, Boston, MA, USA) and the reference genome (accession PRJNA388211). Proteins with a 1.2-fold change and *P*-value ≤ 0.5 between two samples were considered to be significant differentially expressed proteins.

## Figures and Tables

**Figure 1 toxins-11-00092-f001:**
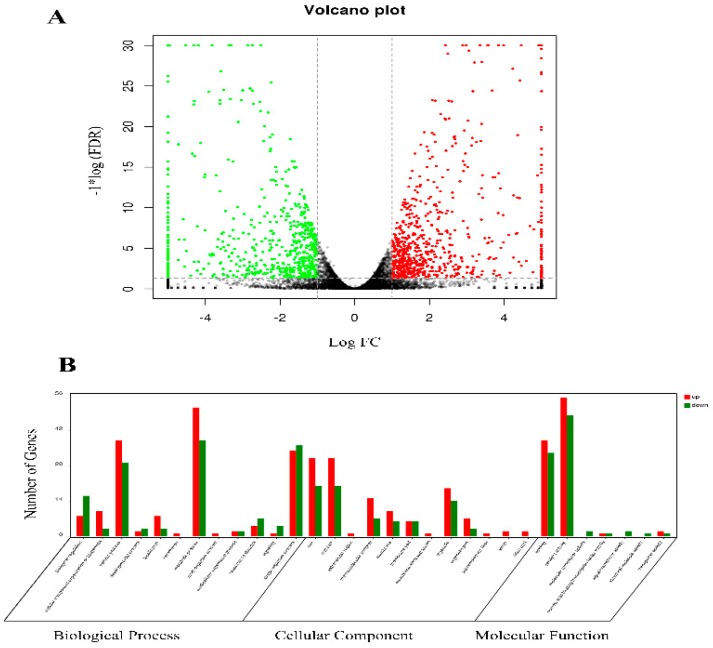
Transcriptomic difference between strain XJ10 and XJ. (**A**) Fold-changes on the *x*-axis represent the ratio of transcript abundance between strain XJ10 and XJ. Differentially expressed transcripts are highlighted in green (down-regulated) and red (up-regulated), respectively on the Volcano plot. (**B**) Gene Ontology (GO) classification of genes differentially expressed between XJ10 and XJ strain and grouped into hierarchically structured GO terms biological process, cellular component, and molecular function.

**Figure 2 toxins-11-00092-f002:**
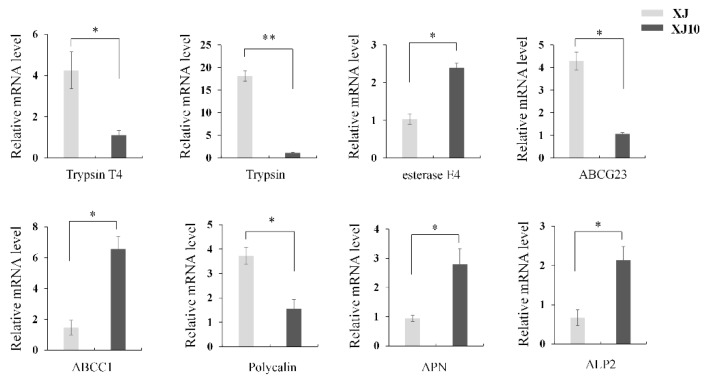
qRT-PCR validation of eight Bt-resistance-related genes differentially expressed between XJ10 and XJ strain. The mRNA levels were compared using Student’s *t*-test (* *P* < 0.05, ** *P* < 0.01).

**Figure 3 toxins-11-00092-f003:**
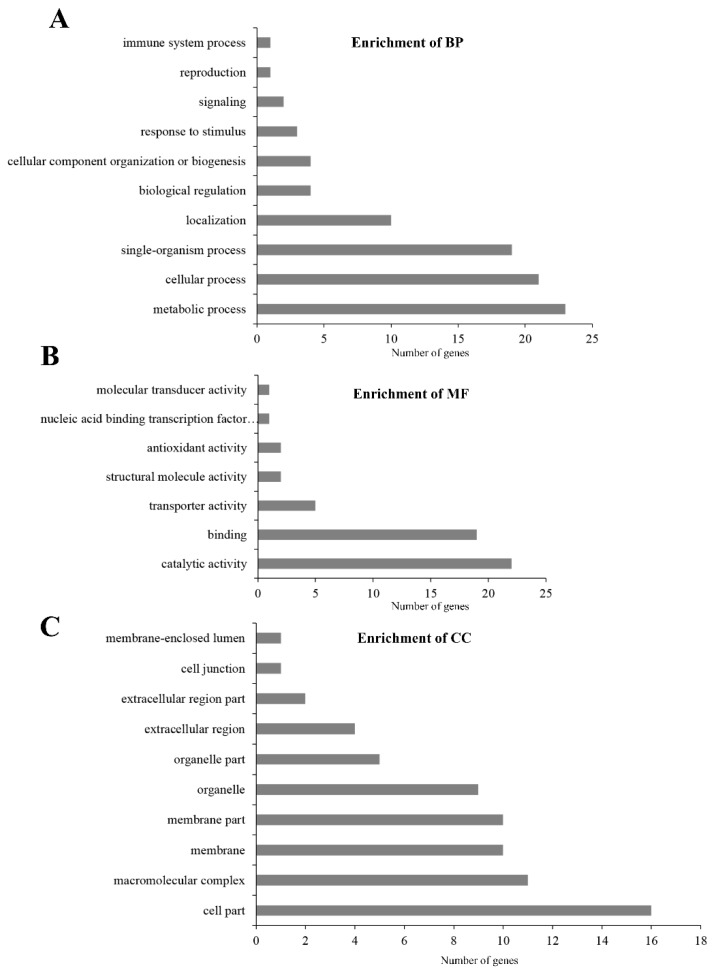
GO analysis of the functional categories of proteins identified from PM. (**A**) Distribution of enriched biological processes (BP). (**B**) Distribution of cell component (CC) enrichment. (**C**) Distribution of molecular function (MF) enrichment.

**Figure 4 toxins-11-00092-f004:**
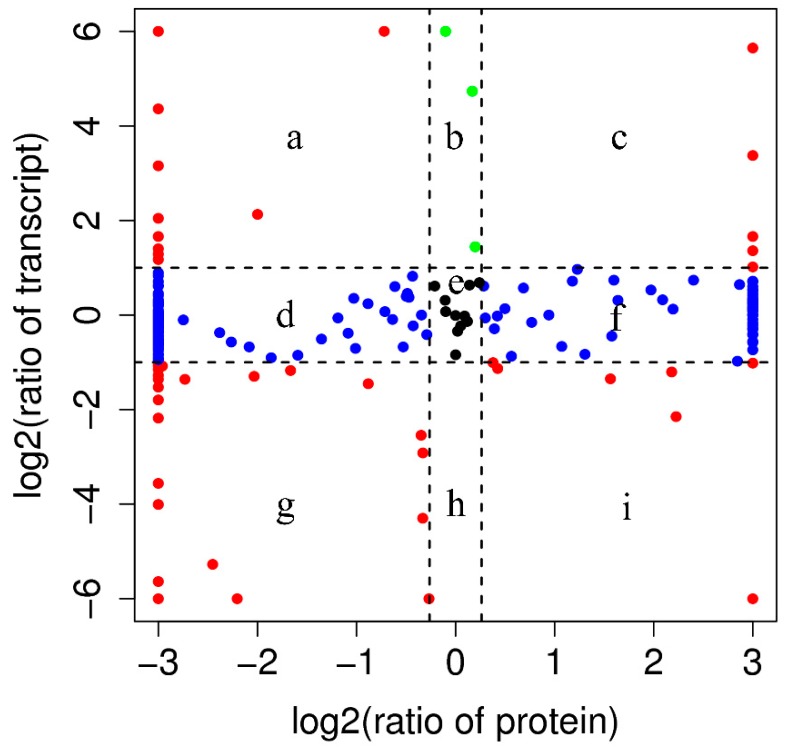
Changes in mRNA and cognate protein abundance between midgut and PM. Relative change in abundance (XJ10/XJ) is shown on a log2 scale. Each letter denotes ratio of abundance of mRNA to protein: e, no significant change in both mRNA and protein; c and g, the expression of mRNA and protein with the same trend; and a, b, d, f, h and i, the expression of mRNA and protein with opposite trends.

**Table 1 toxins-11-00092-t001:** Effects of Cry1Ac protoxin and activated toxin on mortality of *H. armigera* larvae of resistant strain XJ10 and susceptible strain XJ.

Strain	Form of Cry1Ac	LC50 ^a^ (95% Confidence Limits)	Resistance Ratio ^b^
XJ	protoxin	8.41 (6.31–11.25)	1
XJ10	protoxin	1233.91 (923.58–1665.62)	146.79
XJ	activated toxin	7.16 (5.25–9.76)	1
XJ10	activated toxin	322.48 (248.42–417.66)	45.05

^a^ Concentration killing 50% with 95% fiducial limits in ng Cry1Ac per g diet. ^b^ Resistance ratio, LC50 for XJ10 divided by LC50 for XJ.

**Table 2 toxins-11-00092-t002:** Partial list of Bt-resistance-associated genes that are differentially expressed (DEGs) between XJ10 and XJ strain.

Gene ID	FC ^a^	Padj Value	Description	Mechanisms
XM_021337885.1	2.53	1.44 × 10^−6^	trypsin	Altered activation of Cry toxins
XM_021345079.1	1.54	3.60 × 10^−17^	trypsin-like protease	
XM_021337887.1	−2.69	7.99 × 10^−10^	trypsin	
XM_021340602.1	−3.56	5.44 × 10^−9^	trypsin T4	
XM_021340597.1	−4.29	3.58 × 10^−26^	trypsin	
XM_021342117.1	−4.59	6.80 × 10^−11^	trypsin-like proteinase T2α	
XM_021340588.1	−8.89	0.000297197	trypsin	
XM_021345839.1	2.41	6.13 × 10^−24^	chymotrypsin	
XM_021345877.1	1.96	1.30 × 10^−11^	chymotrypsin-like protease	
XM_021334546.1	−5.27	3.40 × 10^−24^	chymotrypsin	
XM_021345783.1	−5.53	3.39 × 10^−67^	chymotrypsin-like protease	
XM_021337546.1	2.03	1.97 × 10^−5^	esterase E4-like	Sequestering the toxin
XM_021334360.1	−1.70	5.59 × 10^−10^	ABC transporter G family 23	ABCs
XM_021345614.1	2.02	8.88 × 10^−19^	ABCC1 protein	
XM_021334935.1	−1.81	1.17 × 10^−11^	Polycalin	Binding proteins
XM_021337081.1	1.66	1.17 × 10^−10^	aminopeptidase N1	
XM_021339316.1	2.24	6.29 × 10^−11^	alkaline phosphatase 2	

^a^ Fold change of DEGs, positive value indicates up-regulation while negative value denotes down-regulation.

**Table 3 toxins-11-00092-t003:** Proteins identified from the peritrophic matrix of *H. armigera.*

Name	Number of Distinct Peptides	MS/MS Number	Sequence Coverage (%)	Predicted MW (kDa)	Accession Number
***Chitin associated***					
mucin 17-like	1	6	0.3	639.67	XM_021332791.1
chitin deacetylase 5a	3	12	7.4	45.297	XM_021341184.1
insect intestinal mucin 2	6	32	7.3	121.82	XM_021326099.1
***Active hydrolases***					
chymotrypsin-like protease C9	5	8	21.1	38.71	XM_021335717.1
trypsin, alkaline C-like	1	1	3.9	27.167	XM_021337856.1
trypsin	1	4	4.3	27.749	XM_021337869.1
trypsin-like protease	2	2	11.9	26.816	XM_021337877.1
trypsin-like protease	2	3	7.5	26.86	XM_021337879.1
trypsin	1	1	6.5	26.217	XM_021338513.1
trypsin-7-like	1	0	2.4	32.302	XM_021340310.1
trypsin T2a	3	8	14.3	27.81	XM_021340589.1
trypsin	1	4	2.7	27.549	XM_021340592.1
trypsin-like protease	5	13	23.9	27.503	XM_021340596.1
trypsin-like protease	7	142	44.9	26.916	XM_021340599.1
trypsin T4	2	6	11	26.772	XM_021340602.1
trypsin 2	1	2	3.3	32.037	XM_021344969.1
chymotrypsin	3	3	14.6	30.518	XM_021345778.1
chymotrypsinogen	6	2	21.4	30.834	XM_021345781.1
chymotrypsin-like protease C8	1	4	6.2	30.106	XM_021345791.1
chymotrypsin-like protease	2	6	13.7	32.376	XM_021345818.1
chymotrypsin-like protease	1	2	8.6	29.92	XM_021345819.1
***carboxypeptidase***					
carboxypeptidase B-like	2	3	4.4	48.983	XM_021328067.1
carboxypeptidase precursor	1	1	1.9	47.903	XM_021330765.1
carboxypeptidase B precursor	2	3	5.1	48.318	XM_021330831.1
carboxypeptidase A	1	2	2.8	48.526	XM_021330834.1
carboxypeptidase	1	1	1.6	42.284	XM_021330844.1
carboxypeptidase	3	5	10.3	47.935	XM_021330848.1
aminopeptidase N	5	14	5	114.37	XM_021337080.1
aminopeptidase N	4	8	4.2	112.81	XM_021337081.1
alpha-amylase	1	1	2	56.009	XM_021332568.1
***Inactive hydrolases***					
serine protease inhibitor 5	1	2	2.8	45.123	XM_021327264.1
serine protease 24	2	2	6.6	43.33	XM_021330937.1
serine protease inhibitor	1	1	1.6	88.323	XM_021334028.1
diverged serine protease	4	13	13.3	27.289	XM_021338518.1
transmembrane protease serine 9-like	2	18	2.5	79.868	XM_021338520.1
serine protease	6	37	45.7	26.952	XM_021340600.1
serine protease inhibitor 3 isoform X1	1	1	1.5	106.56	XM_021343251.1
diverged serine protease	5	13	22.1	25.554	XM_021344422.1
serine protease 52	1	3	4.7	27.001	XM_021344467.1
serine protease 3, partial	3	5	10.7	30.739	XM_021345780.1
lipase	3	9	14.2	30.739	XM_021326676.1
neutral lipase	1	1	3.3	36.909	XM_021331172.1
neutral lipase	1	2	3.6	36.476	XM_021331174.1
neutral lipase	2	5	7.2	36.476	XM_021331175.1
pancreatic lipase 2	2	2	5.7	38.096	XM_021338421.1
lipase	1	1	2.4	36.224	XM_021338439.1
pancreatic lipase 2	2	1	8.7	36.224	XM_021343929.1
neutral lipase	1	2	3.6	35.29	XM_021344621.1
inactive lipase	1	2	5.7	30.895	XM_021344707.1
***Immune-related***					
tubulin alpha-1 chain-like	1	2	2	49.819	XM_021338879.1
***Lipocalins***					
fatty acid-binding protein 3	4	4	33.3	14.744	XM_021333801.1
fatty acid-binding protein 2	2	2	16.4	15.066	XM_021341051.1
fatty acid-binding protein 1	1	1	9.7	14.986	XM_021341061.1
fatty acid-binding protein 2	2	3	4	101.47	XM_021341064.1
polycalin	1	2	0.8	101.47	XM_021334936.1
***Hexamerins***					
arylphorin	8	19	11	82.226	XM_021340131.1
arylphorin	5	4	4.9	82.28	XM_021340132.1
***heat shock protein***					
heat shock protein 90	2	2	1.8	82.63	XM_021341131.1
heat shock protein	3	7	5.5	73.029	XM_021332476.1
*others*					

**Table 4 toxins-11-00092-t004:** Proteins differentially expressed between XJ10 and XJ strain with a fold-change >1.5 and *P*-value < 0.05.

Gene ID	FC ^a^	Padj Value	mRNA FC	Description
XM_021332568.1	21.08226	0.033424	0.24	alpha-amylase 2-like
XM_021341184.1	2.195151	0.03154	0.13	chitin deacetylase 5α
XM_021328621.1	1.576713	0.0048	−0.45	unconventional myosin-XV isoform X2
XM_021337877.1	−1.86068	0.006606	−0.91	trypsin-like protease
XM_021337869.1	−2.20431	0.018165	−7.22	trypsin
XM_021340592.1	−2.73246	0.025215	−1.36	trypsin
XM_021337879.1	−4.27566	0.035587	-	trypsin-like protease
XM_021330834.1	−22.6429	0.000219	1.40	carboxypeptidase A
XM_021340037.1	−23.7458	0.011483	−1.53	uncharacterized protein
XM_021345819.1	−24.6671	0.036741	−0.62	chymotrypsin-like protease
XM_021345818.1	−25.2913	0.032925	−0.02	chymotrypsin-like protease
XM_021340602.1	−28.7418	0.017556	−3.56	trypsin T4

^a^ Fold change of DEGs, positive value indicates up-regulation while negative value denotes down-regulation.

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
