# Peer review of "Comparative Proteomics of Peritrophic Matrix Provides an Insight into its Role in Cry1Ac Resistance of Cotton Bollworm Helicoverpa armigera"

_toxins, 2019, doi:10.3390/toxins11020092_

Round 1

Reviewer 1 Report

Figure 1 is too small to see all the legends (Panel B in particular).

There are a few typos throughout the manuscript.

Author Response

Comments from the reviewers:

Reviewer 1

1. Figure 1 is too small to see all the legends (Panel B in particular).

Thanks for your careful review. As you commented, we have changed the Figure 1 and resubmit it.

2. There are a few typos throughout the manuscript.

Thanks for your reminding and careful review. We invited an English-speaking expert in this field for helping us to edit this manuscript. We have checked carefully through the whole text to correct any grammatical mistakes or expression problems. Thank you again for your valuable and constructive castigation.

Reviewer 2 Report

INTRODUCTION AND CONTEXTUALIZATION:

The presented research concerns an important field, the resistance evolution in target pest insects against Bt toxins. However, I have some fundamental questions that the authors do not address. While they work exclusively with bacterial forms of Bt toxins, protoxins and activated toxins, these forms of Bt toxins play a minor role in agriculture – mostly at least. Microbial Bt-formulations are used primarily in organic production – at times this can be excessive and promote resistance evolution – but mostly Bt sprays are used sparingly, infrequently and on a small area. Hence, resistance evolution against bacterial Bt formulations is typically not what drives this research field. Much rather it is the genetically engineered commodity crops cotton, maize and soybeans that express recombinant, mostly activated versions of the Bt toxins. Since the Bt toxins in these products differ significantly, the important question remains to what degree – or if at all – the research carried out exclusively with bacterial formulations are relevant for the genetically engineered Bt crops where the Bt toxins are produced in significantly different biochemical forms – see Latham et al. for a detailed analyses - Jonathan R. Latham, Madeleine Love & Angelika Hilbeck (2017) The distinct properties of natural and GM cry insecticidal proteins, Biotechnology and Genetic Engineering Reviews, 33:1, 62-96, DOI: 10.1080/02648725.2017.1357295

It is a salient or very casually stated but undocumented assumption that what is found with bacterial forms of Bt toxins will translate one-to-one to Bt crops. At the very least, I think the authors should explicitly address and state their thoughts regarding the relevance of their research, in particular, in light of the documented and undeniable differences between Bt-formulations and Bt-crops.

Furthermore, I find the manuscript contains a wild mix of references, some being very dated – before the days of commercial Bt-crop production (1996) although much more recent literature exists on the topic. For example, on page 1, Introduction, last line, the authors state that ‘the evolution of resistance by pests have become the primary threat to the long-term use of Bt toxins’ and they cite a paper by Tabashnik from 1994(!), 2 years before Bt crops were even commercialized. That resistance has become a primary threat, however, is also certainly not recent – this threat began to undermine the efficacy of Bt crops already over a decade ago when it began to render Bt maize without protection to the target stemborer pest Busseola fusca in South Africa and Bt maize had to sprayed massively with insecticides again 10 years ago until a stacked Bt maize version was introduced. These are old but widely known cases and many more have emerged to date and much more recent literature references on this issue are available. Furthermore, the proposed modes of action have long come under scrutiny not only by Latham et al. 2017, but also by others like Vachon et al. 2012 - J Invertebr Pathol. 2012 Sep 15;111(1):1-12. doi: 10.1016/j.jip.2012.05.001 or yet others who find significant interactions with antibiotic substances (work by Broderick et al. 2006, 2009, Mason et al. 2009 or very recently Hilbeck et al. 2018 - Toxins 2018, 10(12), 489; https://doi.org/10.3390/toxins10120489 ). Also the casual statement that Bt toxins are considered to be free of risk and adverse effects to other organisms (see page 1, Intro, lines 24-25) has come long under scrutiny for both Bt bacterial forms (van Frankenhuyzen 2009, 2013) and Bt crops (e.g. Hilbeck & Otto 2015 - Front. Environ. Sci. https://doi.org/10.3389/fenvs.2015.00071. All in all, the referencing and citation praxis of the authors comes across as somewhat outdated and selective to a certain line of (old) thinking and narrative. This has relevance also for the discussion and the contextualization of their findings within current understanding and debate of Bt toxin modes of action – not only for the targeted pest insects but also for non-target organisms.

MATERIALS, METHODS AND RESULTS: The materials and methods are carefully written and the results are very interesting and certainly publish-worthy – perhaps in broader sense than the authors are realizing as they seem to not be up-to-date regarding the scientific state-of-the-art and debate about nontarget effects of Bt toxins from various sources.

DISCUSSION: As a result of the narrow focus on target pest resistance evolution, the authors may overlook and ignore that the same or similar biological and biochemical processes they observe in their study subjects may be at work in other organisms as well. In particular, I would like to encourage the authors to broaden the horizon and look into some of the reports of Bt effects on nontarget organisms and to what extent their findings may be of relevance to explain or guide other researchers who study the impact of Bt toxins on nontarget organisms to follow their line of research in the quest of understanding what mechanisms may be at work in these nontarget organisms. Certainly, it is mandatory to include some of the cited references above in their discussion and contextualize their results within this on-going and much more recent debate regarding modes of action, interactions with antibiotic substances etc.

The English requires careful copy-editing as there are many small grammatical or other types of mistakes throughout the text. As an example may serve : Page 2, lines 53- 58 – I will indicate the corrections in CAPITAL letters and bold commas: “Our previous work with ….protease activity may BE associated with the …. . Here, we evaluated … of the Cry1Ac protoxin IS much higher THAN the resistance ratio … We hypothesized that this may BE associateD with … take PART in the … Then, we … “

Author Response

Comments from the reviewers:

Reviewer 2

Thank you for the review’s comments concerning our manuscript. Those comments are all valuable and very helpful for revising and improving our paper, as well as the important guiding significance to our researches. We have studied comments carefully and have made correction which we hope meet with approval. Revised portion are marked in yellow in the paper. The main corrections in the paper and the responds to the reviewer’s comments are as flowing:

1. INTRODUCTION AND CONTEXTUALIZATION:

The presented research concerns an important field, the resistance evolution in target pest insects against Bt toxins. However, I have some fundamental questions that the authors do not address. While they work exclusively with bacterial forms of Bt toxins, protoxins and activated toxins, these forms of Bt toxins play a minor role in agriculture – mostly at least. Microbial Bt-formulations are used primarily in organic production – at times this can be excessive and promote resistance evolution – but mostly Bt sprays are used sparingly, infrequently and on a small area. Hence, resistance evolution against bacterial Bt formulations is typically not what drives this research field. Much rather it is the genetically engineered commodity crops cotton, maize and soybeans that express recombinant, mostly activated versions of the Bt toxins. Since the Bt toxins in these products differ significantly, the important question remains to what degree – or if at all – the research carried out exclusively with bacterial formulations are relevant for the genetically engineered Bt crops where the Bt toxins are produced in significantly different biochemical forms – see Latham et al. for a detailed analyses - Jonathan R. Latham, Madeleine Love & Angelika Hilbeck (2017) The distinct properties of natural and GM cry insecticidal proteins, Biotechnology and Genetic Engineering Reviews, 33:1, 62-96, DOI: 10.1080/02648725.2017.1357295

It is a salient or very casually stated but undocumented assumption that what is found with bacterial forms of Bt toxins will translate one-to-one to Bt crops. At the very least, I think the authors should explicitly address and state their thoughts regarding the relevance of their research, in particular, in light of the documented and undeniable differences between Bt-formulations and Bt-crops.

Thanks for your careful review and your professional advice. We absolutely agree with what you said about protoxins and activated toxins play a minor role in agriculture. Bt sprays are also used sparingly and infrequently due to its price and control efficiency. In the main text, we deleted the widely used in sprays (line 22). And we also agree with you that genetically engineered commodity crops drives the resistance evolution rather than Bt sprays. Besides this, we think laboratory-selected resistance strain also play important roles in the understanding of the mode of Bt action. Most of Bt receptors were first discovered in the laboratory. In this study, we also selected Bt-resistance stain under laboratory-conditions. The aim of this study was to compare the difference between strain XJ (susceptible) and XJ10 (resistance) and provide a foundation for resistance research. And we added a sentence to link the purpose of our study to pest control (line 28-29).

2. Furthermore, I find the manuscript contains a wild mix of references, some being very dated – before the days of commercial Bt-crop production (1996) although much more recent literature exists on the topic. For example, on page 1, Introduction, last line, the authors state that ‘the evolution of resistance by pests have become the primary threat to the long-term use of Bt toxins’ and they cite a paper by Tabashnik from 1994(!), 2 years before Bt crops were even commercialized. That resistance has become a primary threat, however, is also certainly not recent – this threat began to undermine the efficacy of Bt crops already over a decade ago when it began to render Bt maize without protection to the target stemborer pest Busseola fusca in South Africa and Bt maize had to sprayed massively with insecticides again 10 years ago until a stacked Bt maize version was introduced. These are old but widely known cases and many more have emerged to date and much more recent literature references on this issue are available. Furthermore, the proposed modes of action have long come under scrutiny not only by Latham et al. 2017, but also by others like Vachon et al. 2012 - J Invertebr Pathol. 2012 Sep 15;111(1):1-12. doi: 10.1016/j.jip.2012.05.001 or yet others who find significant interactions with antibiotic substances (work by Broderick et al. 2006, 2009, Mason et al. 2009 or very recently Hilbeck et al. 2018 - Toxins 2018, 10(12), 489; https://doi.org/10.3390/toxins10120489 ). Also the casual statement that Bt toxins are considered to be free of risk and adverse effects to other organisms (see page 1, Intro, lines 24-25) has come long under scrutiny for both Bt bacterial forms (van Frankenhuyzen 2009, 2013) and Bt crops (e.g. Hilbeck & Otto 2015 - Front. Environ. Sci. https://doi.org/10.3389/fenvs.2015.00071. All in all, the referencing and citation praxis of the  authors comes across as somewhat outdated and selective to a certain line of (old) thinking and narrative. This has relevance also for the  discussion and the contextualization of their findings within current understanding and debate of Bt toxin modes of action – not only for the targeted pest insects but also for non-target organisms.

Thanks for your careful review. We worshiped your erudition and knowledge. And thank you for your reminding, we have added some recent literature references in the main text. Honestly, our research direction focused on the resistance mechanism of lab-selected strain and have little research on non-target organisms. In the future, we will study more about the toxin mode of actions for non-target organisms. Thank you again for your advice.

3. MATERIALS, METHODS AND RESULTS: The materials and methods are carefully written and the results are very interesting and certainly publish-worthy – perhaps in broader sense than the authors are realizing as they seem to not be up-to-date regarding the scientific state-of-the-art and debate about nontarget effects of Bt toxins from various sources.

Thanks for appreciating of our manuscript and careful review. Frankly, we main focus on the target effects of Bt toxins and the resistance mechanism of target pest. We have few studies on non-target effects of Bt toxins. So, we did not discuss the non-target effects of Bt toxins from various sources. But, in the future, we would like to learn more about this.

4. DISCUSSION: As a result of the narrow focus on target pest resistance evolution, the authors may overlook and ignore that the same or similar biological and biochemical processes they observe in their study subjects may be at work in other organisms as well. In particular, I would like to encourage the authors to broaden the horizon and look into some of the reports of Bt effects on nontarget organisms and to what extent their findings may be of relevance to explain or guide other researchers who study the impact of Bt toxins on nontarget organisms to follow their line of research in the quest of understanding what mechanisms may be at work in these nontarget organisms. Certainly, it is mandatory to include some of the cited references above in their discussion and contextualize their results within this on-going and much more recent debate regarding modes of action, interactions with antibiotic substances etc.

Thanks for your careful review and your professional advice. It is really true as you suggested that other organism may have same or similar biological and biochemical processes. Just like, Spodoptera littoralis, Plodia interpunctella, Heliothis virescens and Ostrinia nubilalis, reduced protease activity were also associated with resistance to Cry1Ac toxins. The aim of this study was to elucidate the resistance mechanism of our lab-selected strains and may have few relevance with non-target organism. So, we did not discuss this in the discussion. We appreciate your suggestion which are very helpful to us. In the future, we will study more about the toxin mode of actions for non-target organisms.

5. The English requires careful copy-editing as there are many small grammatical or other types of mistakes throughout the text. As an example may serve : Page 2, lines 53- 58 – I will indicate the corrections in CAPITAL letters and bold commas: “Our previous work with ….protease activity may BE associated with the …. . Here, we evaluated … of the Cry1Ac protoxin IS much higher THAN the resistance ratio … We hypothesized that this may BE associateD with … take PART in the … Then, we … “

Thanks for your reminding and careful review. We invited an English-speaking expert in this field for helping us to edit this manuscript. We have checked carefully through the whole text to correct any grammatical mistakes or expression problems. Thank you again for your valuable and constructive castigation.